# Is Label Smoothing Truly Incompatible with Knowledge Distillation: An Empirical Study

**Zhiqiang Shen**
CMU

**Zechun Liu**
CMU & HKUST

**Dejia Xu**
Peking University

**Zitian Chen**
UMass Amherst

**Kwang-Ting Cheng**
HKUST

**Marios Savvides**
CMU

## Abstract

This work aims to empirically clarify a recently discovered perspective that label smoothing is incompatible with knowledge distillation (Müller et al., 2019). We begin by introducing the motivation behind on how this incompatibility is raised, i.e., *label smoothing erases relative information between teacher logits*. We provide a novel connection on how label smoothing affects distributions of semantically similar and dissimilar classes. Then we propose a metric to quantitatively measure the degree of erased information in sample's representation. After that, we study its one-sidedness and imperfection of the incompatibility view through massive analyses, visualizations and comprehensive experiments on Image Classification, Binary Networks, and Neural Machine Translation. Finally, we broadly discuss several circumstances wherein label smoothing will indeed lose its effectiveness.[1]

## 1 Introduction

Label smoothing (Szegedy et al., 2016) and knowledge distillation (Hinton et al., 2015) are two commonly recognized techniques in training deep neural networks and have been applied in many state-of-the-art models, such as language translation (Vaswani et al., 2017; Tan et al., 2019; Zhou et al., 2020), image classification (Xie et al., 2019; He et al., 2019) and speech recognition (Chiu et al., 2018; Pereyra et al., 2017; Chorowski & Jaitly, 2017). Recently a large body of studies is focusing on exploring the underlying relationships between these two methods, for instance, Müller et al. (Müller et al., 2019) discovered that label smoothing could improve calibration implicitly but will hurt the effectiveness of knowledge distillation. Yuan et al. (Yuan et al., 2019) considered knowledge distillation as a dynamical form of label smoothing as it delivered a regularization effect in training. The recent study (Lukasik et al., 2020) further noticed label smoothing could help mitigate label noise, they showed that when distilling models from noisy data, the teacher with label smoothing is helpful. Despite the massive and intensive researches, how to use label smoothing as well as knowledge distillation in practice is still unclear, divergent, and under-explored. Moreover, it is hard to answer when and why label smoothing works well or not under a variety of discrepant circumstances.

**View of incompatibility between label smoothing and knowledge distillation.** Recently, Müller et al. proposed a new standpoint that *teachers trained with label smoothing distill inferior student compared to teachers trained with hard labels, even label smoothing improves teacher's accuracy*, as the authors found that label smoothing tends to "erase" information contained intra-class across individual examples, which indicates that the relative information between logits will be erased to some extent when the teacher is trained with label smoothing. This rising idea is becoming more and more dominant and has been quoted by a large number of recent literatures (Arani et al., 2019; Tang et al., 2020; Mghabbar & Ratnamogan, 2020; Shen et al., 2020; Khosla et al., 2020).

However, this seems reasonable observation basically has many inconsistencies in practice when adopting knowledge distillation with smoothing trained teachers. Thus, we would like to challenge whether this perspective is entirely correct? To make label smoothing and knowledge distillation less mysterious, in this paper, we first systematically introduce the mechanism and correlation

---

[1]Project page: http://zhiqiangshen.com/projects/LS_and_KD/index.html.

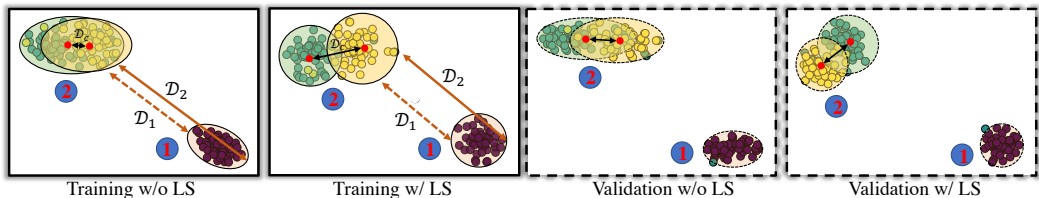

| Training w/o LS | Training w/ LS | Validation w/o LS | Validation w/ LS |

Figure 1: Illustrations of the effects of label smoothing on penultimate layer output. The figure is plotted on ImageNet with ResNet-50 following (Müller et al., 2019), we also choose two semantically similar classes (toy poodle and miniature poodle, in green and yellow) and one semantically different class (tench, in purple). ① is the discovery observed by Müller et al. that label smoothing will enforce each example to be equidistant to its template, i.e., erasing the relative information between logits. $\mathcal{D}_1$ and $\mathcal{D}_2$ are the degree of measuring "how much a tench is similar to poodle". ② is our new finding in this paper that "erasing" effect enabled by label smoothing actually promotes to enlarge relative information on those semantically similar classes, i.e., making them have less overlap on representations. $\mathcal{D}_c$ is the distance between the semantically similar "toy poodle" cluster and the "miniature poodle" cluster. More details can be referred to Sec. 3.

between these two techniques. We then present a novel connection of label smoothing to the idea of "erasing" relative information. We expose the truth that factually the negative effects of erasing relative information only happen on the semantically different classes. Intuitively, those classes are easy to classify as they have obvious discrepancies. Therefore, the negative effects during distillation are fairly moderate. On those semantically similar classes, interestingly, we observe that **erasing phenomenon can enforce two clusters being away from each other and actually enlarge the central distance of clusters between classes**, which means it makes the two categories easier for classifying, as shown in Fig. 1. These classes in traditional training procedure are difficult to distinguish, so generally, the benefits of using label smoothing on teachers outweigh the disadvantages when training in knowledge distillation. Our observation in this paper supplements and consummates prior Müller et al.'s discovery essentially, demonstrates that label smoothing is compatible with knowledge distillation through explaining the erasing logits information on similar classes. We further shed light on understanding the behavior and effects when label smoothing and knowledge distillation are applied simultaneously, making their connection more interpretable, practical and clearer for usage.

**How to prove that their discovery is not judgmatic?** We clarify such widely accepted idea through the following exploratory experiments, and exhaustively evaluate our proposed hypothesis: (i) Standard ImageNet-1K (Deng et al., 2009), fine-grained CUB200-2011 (Wah et al., 2011b) and noisy iMaterialist product recognition; (ii) Binary neural networks (BNNs) (Rastegari et al., 2016); (iii) Neural machine translation (NMT) (Vaswani et al., 2017). Intriguingly, we observe that if the teacher is trained with label smoothing, the absolute values of converged distilling loss on training set are much larger than that the teacher is trained with hard labels, whereas, as we will discuss in detail later in Fig. 5 and 6, the accuracy on validation set is still better than that without label smoothing. We explain this seemingly contradictory phenomenon through visualizing the teachers' output probabilities with and without label smoothing, it suggests that the suppression of label smoothing for knowledge distillation only happens on training phase as the distributions from teachers with label smoothing is more flattening, the generalization ability of networks on validation set is still learned during optimization. That is to say, the dynamical soft labels generated by teacher networks can prevent learning process from overfitting to the training data, meanwhile, improving the generalization on the unseen test data. Therefore, we consider this erasing relative information function within class from label smoothing as a merit to distinguish semantically similar classes for knowledge distillation, rather than a drawback. Moreover, we also propose a stability metric to evaluate the degree of erased information by label smoothing, we found the proposed metric is highly aligned with model's accuracy and can be regarded as a supplement or alternateness to identify good teachers for knowledge distillation. Finally, we discuss several intriguing properties of label smoothing we observed on the long-tailed category distribution and rapidly-increased #class scenarios, as provided in Sec. 7.

More specifically, this paper aims to address the following questions:

● *Does label smoothing in teacher networks suppress the effectiveness of knowledge distillation?* Our answer is *No*. Label smoothing will not impair the predictive performance of students. Instead, we observe that a smoothing trained teacher can protect the student from overfitting on the training set, which means that with smoothing trained teachers in knowledge distillation, the training loss is always higher than that without smoothing, but the validation accuracy is still similar or even better.

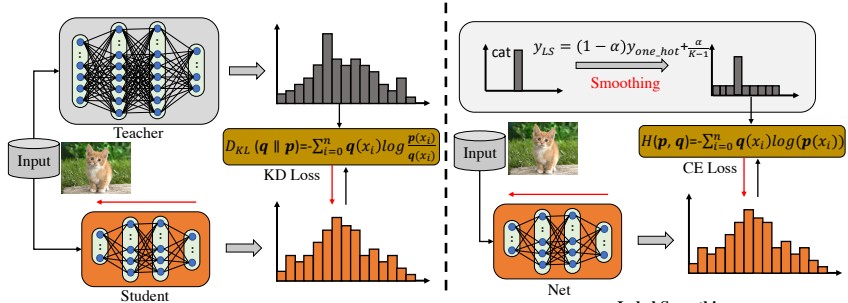

Figure 2: Knowledge distillation (KD) and label smoothing (LS) overview. Both the KD and LS adopt softened distributions for learning the target networks. The KD differs from LS in the generation of these distributions and the objectives for optimization. KD chooses to utilize a pre-trained teacher to produce the supervision dynamically, while LS uses a constant uniform distribution for training. In the figure, the black lines are the forward pass and the red lines are the gradient propagation direction.

● *What will actually determine the performance of a student in knowledge distillation?* From our empirical study, we observe if the student architecture is settled, the dominating factor in knowledge distillation is the quality of supervision, i.e., the performance of a teacher network. A higher-accuracy teacher is particularly successful in distilling a better student, regardless it is trained with or without label smoothing. This observation is partially against the conclusion in (Müller et al., 2019) which stated "a teacher with better accuracy is not necessary to distill a better student".

● *When will the label smoothing indeed lose its effectiveness for learning deep neural networks?* Long-tailed class distribution and increased number of classes are two scenarios we observed wherein label smoothing will lose or impair its effectiveness. We empirically verify the findings on iNaturalist 2019 (Van Horn et al., 2018), Place-LT (Liu et al., 2019) and curated ImageNet (Liu et al., 2019).

## 2 BACKGROUND

In this section, we first introduce the background of label smoothing and knowledge distillation through a mathematical description. Given a dataset $\mathcal{D} = (X, Y)$ over a set of classes $K$, $X$ is the input data and $Y$ is the corresponding one-hot label with each sample's label $\boldsymbol{y} \in \{0, 1\}^K$, where the element $y_c$ is 1 for the ground-truth class and 0 for others. Label smoothing replaces one-hot hard label vector $\boldsymbol{y}$ with a mixture of weighted $\boldsymbol{y}$ and a uniform distribution:

$$y_c = \begin{cases} 1 - \alpha & \text{if } c = label, \\ \alpha/(K-1) & \text{otherwise.} \end{cases} \quad (1)$$

where $\alpha$ is a small constant coefficient for flattening the one-hot labels. Usually, label smoothing is adopted when the loss function is cross-entropy, and the network uses *softmax* function to the last layer's logits $\boldsymbol{z}$ to compute the output probabilities $\boldsymbol{p}$, so the gradient of each training sample with respect to $\boldsymbol{z}$ will be: $\nabla \mathcal{H}(\boldsymbol{p}, \boldsymbol{y}) = \boldsymbol{p} - \boldsymbol{y} = \sum_{c=1}^{K} (\text{Softmax}(z_c) - y_c)$, where $\mathcal{H}(\boldsymbol{p}, \boldsymbol{y}) = - \sum_{c=1}^{K} y_c \log p_c$ is the cross-entropy loss and $z_c$ is $c$-th logit in $\boldsymbol{z}$.

To further understand the effects of label smoothing on loss function, Fig. 3 illustrates correction effects of smoothing on the binary cross-entropy loss ($K = 2$). We can observe that the standard logistic loss ($\alpha = 0$) vanishes for large and confident positive predictions, and becomes linear for large negative predictions. Label smoothing will penalize confident predictions and involve a finite positive minimum as it aims to minimize the average per-class. Generally, larger $\alpha$ values will produce larger loss values rebounding at positive predictions. This is also the underlying reason that smoothed loss can flatten the predictions of a network.

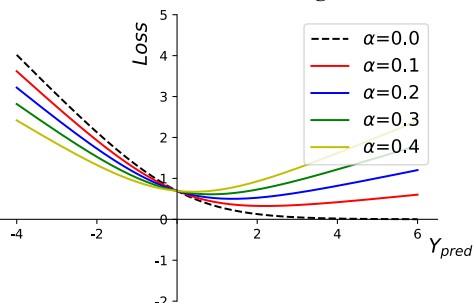

Figure 3: Correction effects of label smoothing on logistic loss with different $\alpha$. Black dotted line presents the standard logistic loss and other colored lines are imposed label smoothing.

In knowledge distillation, we usually pre-train the teacher model $\mathcal{T}_{\mathbf{w}}$ on the dataset in advance. The student model $\mathcal{S}_{\mathbf{w}}$ is trained over the same set of data, but utilizes labels generated by $\mathcal{T}_{\mathbf{w}}$. More specifically, we can regard this process as learning $\mathcal{S}_{\mathbf{w}}$ on a new labeled dataset $\tilde{\mathcal{D}} = (X, \mathcal{T}_{\mathbf{w}}(X))$. Once the teacher network is trained, its parameters will be frozen in the whole distillation.

The student network $\mathcal{S}_{\mathbf{w}}$ is trained by minimizing the similarity between its output and two parts: the hard one-hot labels and the soft labels generated by the teacher network. Letting $p_c^{\mathcal{T}_{\mathbf{w}}}(X) = \mathcal{T}_{\mathbf{w}}(X)[c]$, $p_c^{\mathcal{S}_{\mathbf{w}}}(X) = \mathcal{S}_{\mathbf{w}}(X)[c]$ be the probabilities assigned to class $c$ in the teacher model $\mathcal{T}_{\mathbf{w}}$ and student model $\mathcal{S}_{\mathbf{w}}$. The distillation loss can be formulated as $\lambda\mathcal{H}(\boldsymbol{p}^{\mathcal{S}_{\mathbf{w}}}, \boldsymbol{y}) + (1-\lambda)\mathcal{H}(\boldsymbol{p}^{\mathcal{S}_{\mathbf{w}}}/\mathcal{T}, \boldsymbol{p}^{\mathcal{T}_{\mathbf{w}}}/\mathcal{T})$ where $\mathcal{T}$ is the temperature scaling factor and $\lambda$ is the trade-off coefficient to balance the two terms.

## 3 THE "ERASE INFORMATION" EFFECT BY LABEL SMOOTHING

This section aims to explain the erasing information effect more thoroughly. We start by reproducing the visualization of penultimate layer's activations using the same procedure from (Müller et al., 2019). We adopt ResNet-50 trained with hard and smoothed labels on ImageNet. As shown in Fig. 1, we obtain similar distributions as (Müller et al., 2019). Since examples in training set are the ones used for distillation, we mainly analyze the visualization from the training data. The core finding in (Müller et al., 2019) is that if a teacher is trained with hard labels, representations of examples are distributed in broad clusters, which means that different examples from the same class can have different similarities ($\mathcal{D}_1$ and $\mathcal{D}_2$) to other classes. For a teacher trained with label smoothing, they observed the opposite behavior. Label smoothing encourages examples to lie in tight equally separated clusters, so each example of one class has very similar proximities ($\mathcal{D}_1$ is closer to $\mathcal{D}_2$) to examples of the other classes. Our re-visualization also supports this discovery. The authors derive the conclusion that a teacher with better accuracy is not necessarily to distill a better student. This seems reasonable as the broad clusters can enable different examples from the same class to provide different similarities to other classes, which contains more information for knowledge distillation.

However, if refocusing on the two semantically similar classes, when label smoothing is applied, the clusters are much tighter because label smoothing encourages each example is to be equidistant from all other class's templates, while, the tight cluster substantially promotes different class representations to be separate, i.e., the distance of clusters $\mathcal{D}_c$ increases, which further indicates that different class examples obtain more distinguishable features. This phenomenon is crucial as these difficult classes are the key for boosting classification performance. Generally, it is not necessary to measure "how much a poodle is a particularly similar to a tench" since we have enough evidence to classify them, but it is critical to have information "how different is a toy poodle to a miniature poodle".

**Visualizations of teacher predictions.** We further visualization the mean distribution of different classes crossing examples, as shown in Fig. 4. We average all the probabilities after softmax layer if the examples belong to the same category, and show the first 100 classes in ImageNet. Usually, the probabilities have a major value (the bars in Fig. 4 (1)) that represents model's prediction on category and other small values (i.e., minor predictions in Fig. 4 (2)) indicate that the input image is somewhat similar to those other categories, some discussions about minor predictions are given in Appendix F. Our purpose of this visualization is to make certain of what label smoothing really calibrates in a network and shed light on how it affects the network predictions. We can observe in this figure that a model trained with label smoothing will generate more softened distributions, but the relations across different classes are still preserved. We conjecture the softened supervision is also the reason why teachers with label smoothing produce larger training loss during knowledge distillation. Consequently, *label smoothing will both decrease the variance (verified by following stability metric) and mean predictive values within a class, but will not impair the relations crossing different classes.*

### 3.1 A SIMPLE METRIC FOR MEASURING THE DEGREE OF ERASED INFORMATION

Different from the visualization scheme (Müller et al., 2019) of finding an orthonormal basis of the plane that only studies this problem qualitatively, we further address the "erasing" phenomenon through a statistical metric that is simple yet effective, and can measure the degree of erasing operation quantitatively. Our motivation behind it is straight-forward: If label smoothing erases relative information within a class, the variance of intra-class probabilities will decrease accordingly, thus we can use such variance to monitor the erasing degree, since this metric evaluates the fluctuation

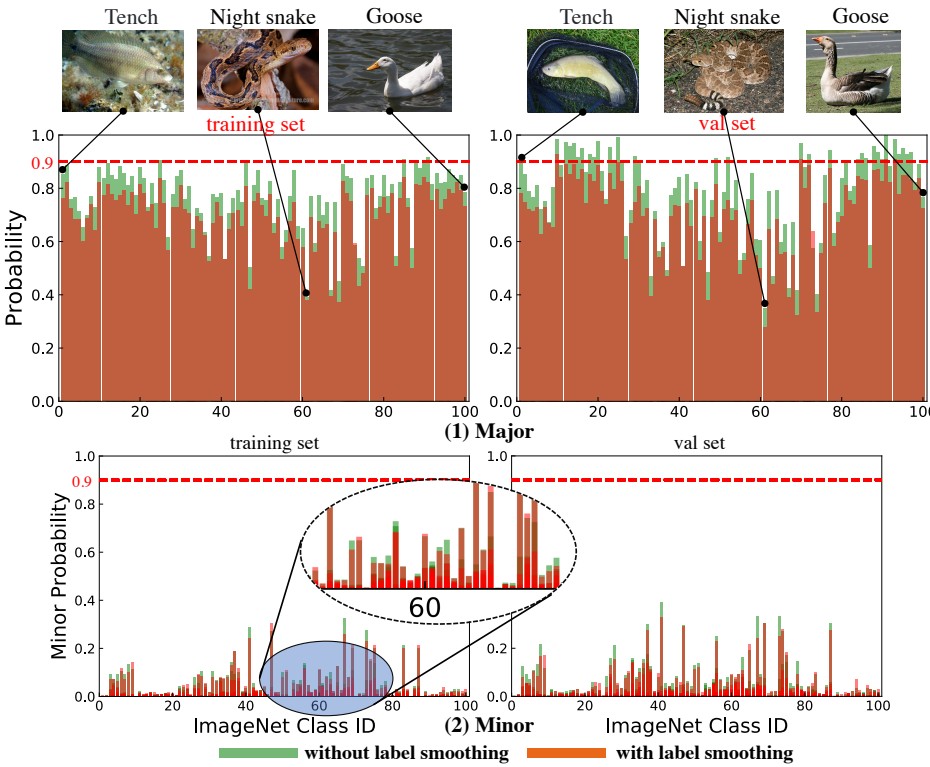

Figure 4: Probability distributions with/without label smoothing on ResNet-50. We show the first 100 categories in ImageNet. The red/green bars are distributions with/without label smoothing, respectively. "Minor probability" denotes the small probabilities predicted by networks when the outputs are used as supervisions in knowledge distillation.

of the representations, we can also call it the stability metric. The definition is as follows:

$$\mathcal{S}_{\text{Stability}} = 1 - \frac{1}{K} \sum_{c=1}^{K} (\frac{1}{\boldsymbol{n}_c} \sum_{i=1}^{\boldsymbol{n}_c} ||\boldsymbol{p}_{\{i,c\}}^{\mathcal{T}_{\mathbf{w}}} - \overline{\boldsymbol{p}}_{\{i,c\}}^{\mathcal{T}_{\mathbf{w}}}||^2) \tag{2}$$

where $i$ is the index of images and $\boldsymbol{n}_c$ is the #image in class $c$. $\overline{\boldsymbol{p}}_{\{i,c\}}^{\mathcal{T}_{\mathbf{w}}}$ is the mean of $p^{\mathcal{T}_{\mathbf{w}}}$ in class $c$. This metric utilizes the probabilities of intra-class variance to measure the stability of a teacher's prediction. The results on various network architectures are shown in Sec. 5 and a PyTorch-like code for calculating this metric is given in Appendix C.

Such metric has at least two advantages: 1) It can measure the degree of erased information quantitatively and further help discover more interesting phenomena, e.g., we observe that data augmentation method like CutMix (Yun et al., 2019) together with longer training erases the relative information on logits dramatically and can further be reinforced by label smoothing. 2) We found that the proposed metric is highly aligned with model accuracy, thus such metric can be used as a complement for accuracy to evaluate the quality of teacher's supervision for knowledge distillation.

## 4 A CLOSE LOOK AT LABEL SMOOTHING AND KNOWLEDGE DISTILLATION

A few recent studies (Shen & Savvides, 2020; Shen et al., 2019) suggested supervised part $\mathcal{H}(\boldsymbol{p}^{\mathcal{S}_{\mathbf{w}}}, \boldsymbol{y})$ (i.e. hard labels) is not necessary as soft prediction is adequate to provide crucial information for students, meanwhile, removing supervised part can avoid involving incorrect labels caused by random crop, multi-object circumstance or false annotations by humans. Thus, here we only consider the soft part $\mathcal{H}(\boldsymbol{p}^{\mathcal{S}_{\mathbf{w}}}/\mathcal{T}, \boldsymbol{p}^{\mathcal{T}_{\mathbf{w}}}/\mathcal{T})$ with the commonly used *Kullback-Leibler divergence* similarity.
**KL-divergence** measures the similarity of two probability distributions. We train the student network $\mathcal{S}_\theta$ by minimizing the KL-divergence between its output $p_c^{\mathcal{S}_\theta}(X)$ and the soft labels $p_c^{\mathcal{T}_\theta}(X)$ generated by the teacher network. Following (Müller et al., 2019; Hinton et al., 2015) we set $\mathcal{T} = 1$ as the

temperature constant and it is omitted for simplicity, thus our loss function will be:

$$\mathcal{D}_{KL}(\mathcal{T}_{\mathbf{w}}\|\mathcal{S}_{\mathbf{w}}) = \mathbb{E}_{x \sim \mathcal{T}_{\mathbf{w}}}\left[-\log \frac{\mathcal{S}_{\mathbf{w}}(X)}{\mathcal{T}_{\mathbf{w}}(X)}\right] \tag{3}$$
$$= \mathbb{E}_{x \sim \mathcal{T}_{\mathbf{w}}}[-\log \mathcal{S}_{\mathbf{w}}(X)] - \mathcal{H}(\mathcal{T}_{\mathbf{w}}(X))$$

Here, $\mathbb{E}_{x \sim \mathcal{T}_{\mathbf{w}}}[-\log \mathcal{S}_{\mathbf{w}}(X)]$ is the cross-entropy between $\mathcal{S}_{\mathbf{w}}$ and $\mathcal{T}_{\mathbf{w}}$ (denoted $\mathcal{H}(\boldsymbol{p}^{\mathcal{S}_{\mathbf{w}}}, \boldsymbol{p}^{\mathcal{T}_{\mathbf{w}}})$). The second term $\mathcal{H}(\mathcal{T}_{\mathbf{w}}(X)) = \mathbb{E}_{x \sim \mathcal{T}_{\mathbf{w}}}[-\log \boldsymbol{p}^{\mathcal{T}_{\mathbf{w}}}(x)]$ is the entropy of teacher $\mathcal{T}_{\mathbf{w}}$ and is constant with respect to $\mathcal{T}_{\mathbf{w}}$. We can remove it and simply minimize the loss as follows:

$$\mathcal{H}(\boldsymbol{p}^{\mathcal{S}_{\mathbf{w}}}, \boldsymbol{p}^{\mathcal{T}_{\mathbf{w}}}) = -\sum_{c=1}^{K} p_c^{\mathcal{T}_{\mathbf{w}}}(X)\log p_c^{\mathcal{S}_{\mathbf{w}}}(X). \tag{4}$$

We can observe that Eq. 4 is actually a standard cross-entropy loss. Then, we have:

**Property 1.** *If not consider hard labels in knowledge distillation, distillation loss and cross-entropy loss with label smoothing have the same optimizing objective, i.e., $\mathcal{D}_{KL}(\mathcal{T}_{\mathbf{w}}\|\mathcal{S}_{\mathbf{w}}) = \mathcal{H}(\boldsymbol{p}^{\mathcal{S}_{\mathbf{w}}}, \boldsymbol{p}^{\mathcal{T}_{\mathbf{w}}})$.*

This property shows that label smoothing and knowledge distillation have the same optimization objective, the sole difference between them is the mechanism of producing the soft labels. Therefore, except for the neural machine translation, in this paper all of our knowledge distillation experiments are conducted without the hard labels, which means our student solely relies on the softened distribution from a teacher without the one-hot ground-truth. This may challenge the common practice in knowledge distillation (Hinton et al., 2015; Romero et al., 2015) that adopted both hard and soft labels with the cross-entropy loss for distillation, while our surprisingly good results and previous studies (Shen & Savvides, 2020; Shen et al., 2019; Bagherinezhad et al., 2018) indicate that knowledge distillation enabled by soft labels solely is not only an auxiliary regularization (Yuan et al., 2019) but can be the dominating supervisions, which further inspires us to carefully revisit the role of knowledge distillation and design better supervision/objective in training deep neural networks.

## 5 EMPIRICAL STUDIES

**Metric Evaluation.** Our results of stability metric are shown in Table 1, the second and third columns are results without label smoothing and the last two are with it. We study the metric crossing a variety of different network architectures. The gaps of $\mathcal{S}_{\mathbf{Stability}}$ using the same architecture measure the degree of erasing relative information. We can observe that the variances (1-$\mathcal{S}_{\mathbf{Stability}}$) with label smoothing always have lower values than models trained without label smoothing, this proves that label smoothing will erase information and enforce intra-class representations of samples being similar. Generally, the accuracy and stability have a positive correlation between them. But the stability can even overcome some outliers, for example, Wide ResNet50 with label smoothing has lower accuracy, but the stability is still consistent with the tendency of predictive quality. Moreover, models trained with more epochs and augmentation techniques like CutMix (Yun et al., 2019) can dramatically increase the stability, this means relative information will be erased significantly by more augmentation together with longer training. We emphasize that this discovery cannot be observed by the qualitative visualization method (Müller et al., 2019). A PyTorch-like code is in Appendix C.

Table 1: Accuracy and stability results with and without label smoothing on ImageNet-1K. Here we show (1-$\mathcal{S}_{\mathbf{Stability}}$), which denotes the aggregated intra-class variance (the lower the better). Red numbers are the quantitative values of the erased information by label smoothing.

| Netowrks | Acc. (%) w/o LS | (1-$\mathcal{S}_{\mathbf{Stability}}$) w/o LS | Acc. (%) w/ LS | (1-$\mathcal{S}_{\mathbf{Stability}}$) w/ LS |
|---|---|---|---|---|
| ResNet-18 (He et al., 2016) | 69.758/89.078 | 0.3359 | **69.774/89.122** | **0.3358 (-0.0001)** |
| ResNet-50 (He et al., 2016) | 75.888/92.642 | 0.3217 | **76.130/92.936** | **0.3106 (-0.0111)** |
| ResNet-101 (He et al., 2016) | 77.374/93.546 | 0.3185 | **77.726/93.830** | **0.3070 (-0.0115)** |
| MobileNet v2 (Sandler et al., 2018) | 71.878/90.286 | 0.3341 | – | – |
| DenseNet121 (Huang et al., 2017) | 74.434/91.972 | 0.3243 | – | – |
| ResNeXt50 32×4d (Xie et al., 2017) | 77.618/93.698 | 0.3229 | **77.774/93.642** | **0.3182 (-0.0047)** |
| Wide ResNet50 (Zagoruyko & Komodakis, 2016) | **78.468/94.086** | 0.3201 | 77.808/93.682 | 0.3155 (-0.0046) |
| ResNeXt101 32×8d (Xie et al., 2017) | 79.312/94.526 | 0.3177 | **79.698/94.768** | **0.3116 (-0.0061)** |
| ResNet50+Long | 76.526/93.070 | 0.3222 | **77.106/93.340** | **0.3090 (-0.0132)** |
| ResNet50+Long+CutMix (Yun et al., 2019) | 76.874/93.500 | 0.2999 | **77.274/93.304** | **0.2890 (-0.0109)** |

**Image Classification.** We verify our perspective through investigating the effectiveness of knowledge distillation with label smoothing on the image classification tasks. We conduct experiments on three

Table 2: Image classification results on ImageNet-1K, CUB200-2011 and iMaterialist product recognition (in Appendix D). The teacher networks with label smoothing are denoted by "✔". We report average over 3 runs for all the teacher network training and student distillation.

| ImageNet-1K (Standard): | | | | |
|---|---|---|---|---|
| Teacher | w/ LS | Acc. (Top1/Top5) | Student | Acc. (Top1/Top5) |
| ResNet-50 | ✘ | $76.056 \pm 0.119/92.791 \pm 0.106$ | ResNet-18 | $71.425 \pm 0.038/90.185 \pm 0.075$ |
| | | | ResNet-50 | $76.325 \pm 0.068/92.984 \pm 0.043$ |
| | ✔ | $\mathbf{76.128 \pm 0.069/92.977 \pm 0.030}$ | ResNet-18 | $\mathbf{71.816 \pm 0.017/90.466 \pm 0.074}$ |
| | | | ResNet-50 | $\mathbf{77.052 \pm 0.030/93.376 \pm 0.015}$ |

| CUB200-2011 (Fine-grained): | | | | |
|---|---|---|---|---|
| Teacher | w/ LS | Acc. (Top1/Top5) | Student | Acc. (Top1/Top5) |
| ResNet-50 | ✘ | $79.931 \pm 0.037/94.370 \pm 0.064$ | ResNet-18 | $77.116 \pm 0.086/93.241 \pm 0.108$ |
| | | | ResNet-50 | $80.910 \pm 0.033/94.738 \pm 0.114$ |
| | ✔ | $\mathbf{81.497 \pm 0.035/95.043 \pm 0.112}$ | ResNet-18 | $\mathbf{78.382 \pm 0.099/93.621 \pm 0.120}$ |
| | | | ResNet-50 | $\mathbf{82.355 \pm 0.050/95.440 \pm 0.075}$ |

datasets: ImageNet-1K (Deng et al., 2009), CUB200-2011 (Wah et al., 2011a) and iMaterialist product recognition challenge data (in Appendix D). We adopt ResNet-{50/101} as teacher networks and ResNet-{18/50/101} as students, respectively. More experimental settings are in Appendix A.

**Results.** The visualizations of our distillation training and testing curves are shown in Fig. 5. A more detailed comparison is listed in Tables 2 and 7. From the visualization we found two interesting phenomena: On training set, the loss of teacher networks that trained with label smoothing is much higher than that of without label smoothing. While on validation set the accuracy is comparable or even slightly better (The boosts on CUB is greater than those on ImageNet-1K, as shown in Table 2). We also provide the experiments of combining hard and soft labels in Appendix E, the results still show the effectiveness of better teachers with label smoothing, which distill better students. To make it clearer why this happens in distillation, we visualize the supervisions from teacher networks in Fig. 4 and the discussion is shown there. It indicates that label smoothing flattens teacher's predictions which causes the enlarged training loss, while the student's generalization ability is still preserved.

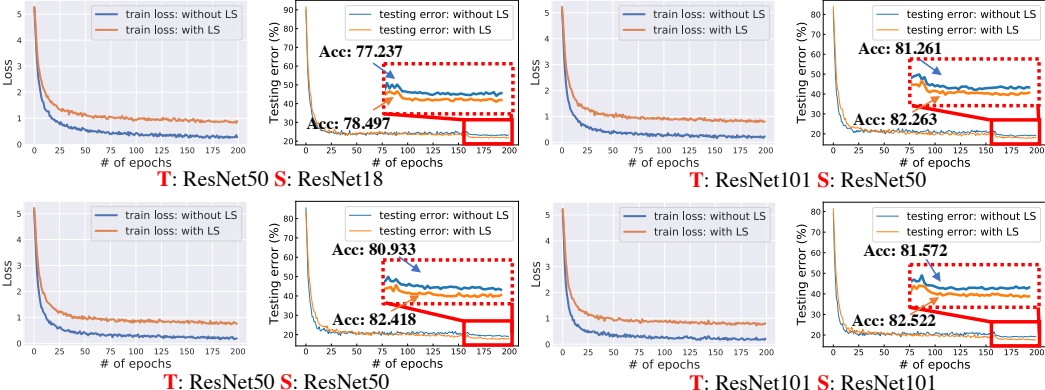

Figure 5: The training and testing curves of knowledge distillation on CUB200-2011 when teachers are trained w/ and w/o label smoothing. The specific teacher and student architectures are given below each subfigure, therein, T indicates the teacher architecture and S indicates the student.

**Binary Neural Networks (BNNs).** We then examine the effectiveness of knowledge distillation on Binary Neural Networks. BNN aims to learn a network that both weights and activations are discrete values in {-1, +1}. In the forward pass, real-valued activations are binarized by the sign function: $\mathcal{A}_b = \text{Sign}(\mathcal{A}_r) = \begin{cases} -1 & \text{if } \mathcal{A}_r < 0, \\ +1 & \text{otherwise.} \end{cases}$ where $\mathcal{A}_r$ is the real-valued activation of the previous layers, produced by the binary or real-valued convolution operations. $\mathcal{A}_b$ is the binarized activation.

The real-valued weights are binarized by: $\mathbf{W}_b = \frac{||\mathbf{W}_r||_{l_1}}{n}\text{Sign}(\mathbf{W}_r) = \begin{cases} -\frac{||\mathbf{W}_r||_{l_1}}{n} & \text{if } \mathbf{W}_r < 0, \\ +\frac{||\mathbf{W}_r||_{l_1}}{n} & \text{otherwise.} \end{cases}$

where $\mathbf{W}_r$ is the real-valued weights that are stored as *latent* parameters to accumulate the small gradients. $\mathbf{W}_b$ is the binarized weights. We update binary weights through multiplying the sign of real-valued latent weights and the channel-wise absolute mean ($\frac{1}{n}||\mathbf{W}_r||_{l_1}$). Training BNNs is challenging as the gradient of optimization is approximated and the capacity of models is also limited.

We perform experiments on ImageNet-1K and results are shown in Fig. 6. Withal, the teacher network trained with one-hot labels (blue curve) is over-confident as the loss value is much smaller, which

means that the teacher trained with label smoothing can prevent distillation process from being over-confident on the training data, and obtain slightly better generalization and accuracy (63.108% *vs.* 63.002%) on the validation set. These results still support our conclusion on knowledge distillation.

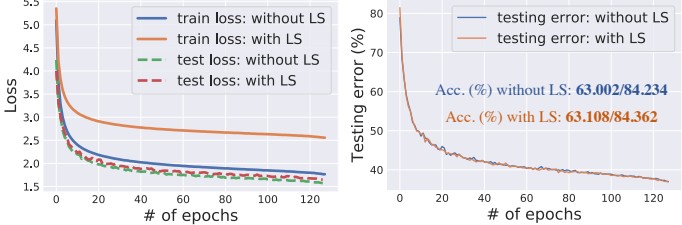

Figure 6: Left is the averaged train/test loss curves in distillation, right is the testing error w/ best Top-1/5 accuracy. We use linear learning rate decay following other binary network training protocol (Martinez et al., 2020; Liu et al., 2018). Our teacher networks are ResNet-50 with and without label smoothing which have similar performance. The student is the state-of-the-art ReActNet (Liu et al., 2020) with ResNet-18 backbone. We can observe that when the teacher is trained with label smoothing, the distillation loss is much higher, but the accuracy of student is still better.

**Neural Machine Translation (NMT).** Finally, we investigate our hypothesis of knowledge distillation on the English-to-German translation task using the Transformer architecture (Vaswani et al., 2017). We utilize the distillation framework of (Tan et al., 2019) on IWSLT dataset, and the pre-training/distillation curves are shown in Fig. 7. A consistent setting is imposed on all the two comparison experiments, except the teacher is trained with and without label smoothing. We choose $\alpha = 0.1$ for label smoothing as suggested by (Vaswani et al., 2017; Szegedy et al., 2016; Müller et al., 2019), we use Adam (Kingma & Ba, 2014) as the optimizer, *lr* with 0.0005, dropout

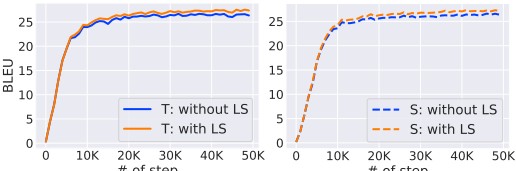

Figure 7: Illustrations of BLEU score curves for teacher pre-training and student distillation. The left figure is teachers' pre-training with and without label smoothing. The right one is the distillation process of students.

with drop rate as 0.3, weight-decay with 0 and max tokens with 4096, all of these hyper-parameters are following the original settings of (Tan et al., 2019). Our results of Fig. 7 deliver two important conclusions: First sub-illustration (left one) proves the statement of Vaswani et al. (Vaswani et al., 2017) that label smoothing ($\alpha = 0.1$) boosts the BLEU score of language model despite causing worse perplexity if comparing to a model is trained with one-hot/hard labels. Second sub-illustration (right one) implies that on the machine translation task, a stronger teacher (trained with label smoothing) will still distill a higher BLEU student. That is to say, label smoothing may not suppress the effectiveness of knowledge distillation in the NMT task.

## 6 WHAT IS A BETTER TEACHER IN KNOWLEDGE DISTILLATION?

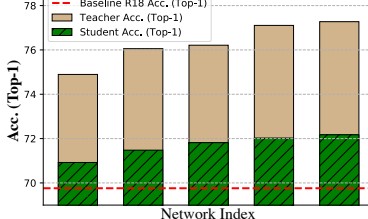
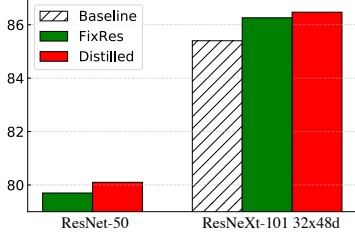

Figure 8: Left is the accuracy relationship between teachers and students, wherein, all teachers are trained with label smoothing. Right is the accuracy of knowledge distillation by using strong teacher to fine-tune the student, FixRes (Touvron et al., 2019) is adopted in both teacher and student networks.

**Better Supervision is Crucial for Distillation.** We further explore the effects of teacher's accuracy on the student through fixing the student structure and switching different teachers. We perform two settings for this ablation study: using the same teacher structure with different training strategies and different teacher architectures. All teacher models are re-trained with label smoothing. The results are shown in Fig. 8 (Left) and Table 8, generally, teachers with higher accuracies can distill stronger students, but they are not linear related and are limited to the capability of the student itself. To further

support the argument that *better teachers usually distill better students*, we choose the state-of-the-art FixRes model (Touvron et al., 2019) for both the teacher and student and perform our distillation training via Eq. 4. The results are shown in Fig. 8 (Right) and our method is slightly better than the baseline and FixRes. Note that the compared FixRes is already the state-of-the-art with ResNet and ResNeXt architecture, so our result (under ResNet family) is a fairly competitive single-crop accuracy to date on ImageNet-1K.

## 7   WHAT CIRCUMSTANCES INDEED WILL MAKE LS LESS EFFECTIVE?

**Long-Tailed Distribution.** We found the long-tailed datasets with imbalanced samples across classes will indeed suppress the effectiveness of label smoothing (LS) and hurt model performance. The results on long-tailed ImageNet-LT, Places365-LT and iNaturalist 2019 are shown in Table 3. Our results on these three datasets empirically verify this observation and support this conclusion. Since LS has weight shrinkage and regularisation effects (Lukasik et al., 2020), we have following conjecture:

**Conjecture 1.** *Weight shrinkage effect (Regularisation) enabled by label smoothing is no longer effective on the long-tailed recognition circumstance and will further impair the performance.*

The weight shrinkage effect (label smoothing regularization) has been proven in (Lukasik et al., 2020) (Theorem 1) on the linear models. As label smoothing will assign probabilities equally to all minor classes (i.e., $(1 - \alpha) \cdot \mathbf{I}_{pos} + \frac{\alpha}{L-1} \cdot \mathbf{J}_{neg}$), this operation may be biased to many-shot classes in the long-tailed scenario. We derive the conclusion empirically that label smoothing is inapplicable when the class distribution is long-tailed.

**More #Class.** This is another circumstance we found will impair the effectiveness of label smoothing. The results are shown in Table 4 and Fig. 9, we average the gains of label smoothing across two different network architectures and compare the boosts between curated ImageNet-100/500 and 1K. Generally, more classes will reduce the improvement produced by label smoothing.

Table 3: Teacher results on the long-tailed ImageNet-LT, Places365-LT and iNaturalist 2019 val set.

| Teacher | w/ LS | ImageNet-LT Acc. (Top1/Top5) | Place-LT Acc. (Top1/Top3) | iNaturalist 2019 Acc. (Top1/Top3) |
|---|---|---|---|---|
| ResNet-18 | ✗ | **39.975/64.645** | **26.479/47.233** | **67.195/83.465** |
|  | ✔ | 39.115/63.655 | 25.877/46.260 | 66.700/83.432 |
| ResNet-34 | ✗ | **41.150/66.205** | **27.329/48.753** | **70.165/86.304** |
|  | ✔ | 40.965/65.850 | 26.863/48.110 | 69.406/86.073 |
| ResNet-50 | ✗ | **40.985/66.030** | 27.384/**48.740** | **73.729/88.845** |
|  | ✔ | 39.965/65.195 | **27.562**/47.945 | 72.904/87.954 |
| ResNet-101 | ✗ | –/– | **28.096/50.164** | **74.389**/88.416 |
|  | ✔ | –/– | 27.466/48.781 | 73.597/**88.779** |

Table 4: Teacher results on the curated ImageNet dataset when increasing the number of classes.

| Teacher | w/ LS | ImageNet-100 Acc. (Top1/Top5) | ImageNet-500 Acc. (Top1/Top5) | ImageNet-1K Acc. (Top1/Top5) |
|---|---|---|---|---|
| ResNet-18 | ✗ | 82.380/**95.520** | 73.521/91.642 | **69.758**/89.076 |
|  | ✔ | **82.740**/95.440 | **74.123/92.004** | 69.606/**89.372** |
| ResNet-101 | ✗ | 82.000/94.340 | 81.712/95.080 | 77.374/93.546 |
|  | ✔ | **83.400/95.300** | **82.020/95.300** | **77.836/93.662** |
| Average (↑) |  | ↑**0.880/0.440** | ↑0.455/0.291 | ↑0.155/0.206 |

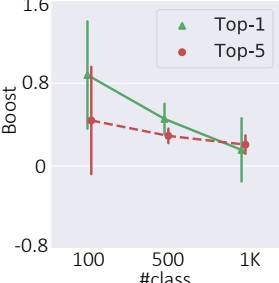

Figure 9: Acc. downtrend.

## 8   CONCLUSION

We empirically demonstrated that label smoothing could both decrease the variance (i.e., erase relative information between logits) and lower mean predictive values (i.e., make prediction less confident) within a category, but it does not impair the relation distribution across different categories. Our results on image classification, binary neural networks, and neural machine translation indicate that label smoothing is compatible with knowledge distillation and this finding encourages more careful to understand and utilize the relationships of label smoothing and knowledge distillation in practice. We found through extensive experiments and analyses that the indeed circumstances label smoothing will lose its effectiveness are long-tailed distribution and increased number of classes. Our study also suggests that, to find a better teacher for knowledge distillation, accuracy of teacher network is one factor, the stability of supervision from teacher network is also an alternative indicator.

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

# APPENDIX

In this appendix, we provide details omitted in the main text, including:

• Section A: A introduction of datasets and experimental settings (Sec. 5 "Empirical Studies" of the main paper.)

• Section B: A discussion about what happens if teachers with label smoothing are inferior? (Sec. 6 "What Is a Better Teacher in Knowledge Distillation?" of the main paper.)

• Section C: A PyTorch-like code for calculating stability metric (Sec. 3.1 "A Simple Metric" of the main paper.)

• Section D: (1) Table 7 is a detailed version of our Table 2 (Sec. 5 "Empirical Studies" of the main paper.) (2) Table 8 is a detailed results for visualization of Fig. 8 (Sec. 6 "What Is a Better Teacher in Knowledge Distillation?" of the main paper.)

• Section E: Results of knowledge distillation by adopting both hard and soft labels (Sec. 5 "Empirical Studies" of the main paper.)

• Section F: Discussions of minor probabilities from teacher network. (Sec. 3 "The Erase Information Effect by Label Smoothing" of the main paper.)

• Section G: Supplementary metric of inter-class variation. (Sec. 5 "Metric Evaluation" of the main paper.)

## A   DATASETS AND EXPERIMENTAL SETTINGS

### A.1   FOR TRAINING TEACHER AND KNOWLEDGE DISTILLATION MODELS TO VERIFY THE COMPATIBILITY BETWEEN LABEL SMOOTHING AND KNOWLEDGE DISTILLATION

**Standard ImageNet-1K Classification (Deng et al., 2009)** ImageNet-1K contains ∼1.28 million images in 1000 classes for training and 50K images for validation. For training teacher networks, we follow the standard training protocol (He et al., 2016; Goyal et al., 2017), i.e., total training epoch is 90, initial learning rate is 0.1 and decayed to 1/10 with every 30 epochs. For distillation, as the supervision is a soft distribution and will dynamically change, we train with 200 epochs and the learning rate is multiplied by 0.1 at 80 and 160 epochs. All models are trained from scratch.

**Fine-grained Recognition on CUB200-2011 (Wah et al., 2011a)** This dataset contains 200 bird species and 11,788 images, and is a wildly-used fine-grained classification benchmark. Following (Wah et al., 2011a), we use standard split with ∼30 samples of each breed for both training and testing. As CUB200-2011 has limited training data, we fine-tune our model from the ImageNet pre-trained parameters. Both teachers and students are trained with 200 epochs and the learning rate is multiplied by 0.1 at 80 and 160 epochs.

**Noisy Product Recognition (iMaterialist Challenge)[2].** The training data consists of 1,011,532 images from 2,019 categories. The validation data has 10,095 images (around 5 for each category). The training data is collected from the Internet and contains some noise (∼30% of incorrect labels), the validation data has been cleaned by human annotators. The training protocol is following CUB200-2011. While, as the training images in this dataset is sufficiently enough, all our models are trained from scratch.

### A.2   FOR TRAINING TEACHERS ONLY TO EXPLORE THE EFFECTIVENESS OF LABEL SMOOTHING

**Long-tailed iNaturalist 2019 (Van Horn et al., 2018).** The iNaturalist Challenge 2019 dataset contains 1,010 species, with a combined training and validation set of 268,243 images, wherein, the validation set has a uniform distribution of three images in each category. While, the training set is constructed with a long-tailed distribution of image numbers. We conduct experiments using the

---

[2]https://sites.google.com/view/fgvc6/competitions/imat-product-2019?authuser=0.

same training protocol as the CUB200-2011 dataset. we evaluate our results on validation set using Top-1/Top-3 metrics. Also, this protocol is applied to the Place-LT dataset.

**Place-LT (Liu et al., 2019).** The training set of this dataset is constructed by a Pareto distribution (Reed, 2001) with a power value $\alpha = 6$. It contains a total number of 184.5K images from 365 categories. There are 4,980~5 images per class from the 365 classes of Places-365 dataset (Zhou et al., 2017). 20 images per class are randomly selected from the original Places-365 training set as a validation set. The training protocol is the same as iNaturalist 2019, we also evaluate our results on validation set using Top-1/Top-3 metrics.

**ImageNet-LT (Liu et al., 2019).** The long-tailed ImageNet is constructed by sampling a subset from the standard ImageNet (Deng et al., 2009) following the Pareto distribution with the power value $\alpha = 6$. The whole dataset contains 115.8K images from 1000 categories, with maximal 1280 images and minimal 5 images per class. 20 training images per class are randomly selected from the origin training set of ImageNet as a validation set. All our models are trained from scratch.

**Curated ImageNet-100/500/1K.** We construct this dataset by randomly selecting a subset of 100/500/1K categories in the standard ImageNet (Deng et al., 2009). We use this dataset to explore the impact of #class for label smoothing. We observe that label smoothing is more effective on the dataset with fewer numbers of classes.

Table 5: Overview of six datasets used in our experiments. $\alpha$ indicates the Pareto distribution value.

| Dataset | #class | Property | Long-tailed $\alpha$ | Head class size | Tail class size | #Training set | #Testing set |
|---|---|---|---|---|---|---|---|
| ImageNet-1K | 1000 | Standard | N/A | N/A | N/A | 1,281,167 | 50,000 |
| CUB200-2011 | 200 | Fine-grained | N/A | N/A | N/A | 5,994 | 5,794 |
| iMaterialist | 2019 | Noisy (~30%) | N/A | N/A | N/A | ~1M | ~10K |
| iNaturalist 2019 | 1010 | Long-tailed | N/A | ~250 | ~210 | 265,213 | 3,030 |
| ImageNet-LT | 1000 | Long-tailed | 6 | 389 | 611 | 115,846 | 20,000 |
| Place-LT | 365 | Long-tailed | 6 | ~100 | ~100 | 184,500 | 7,300 |

## B   WHAT HAPPENS IF TEACHERS WITH LABEL SMOOTHING ARE INFERIOR?

In the main paper, we discussed that the superior teachers (even trained with label smoothing) could distill better students. Also, as we noted that long-tailed distribution will suppress the effectiveness of label smoothing. In this section, we would like to explore what happens if we use these inferior teachers for knowledge distillation? We conduct experiments on the long-tailed iNaturalist 2019 data and show more evidence to prove our conclusion that the quality of supervision is more crucial for knowledge distillation than the ways of training teachers (with or without label smoothing). The results are shown in Table 6, it can be observed that if teachers with lower accuracies, the distilled students will also have poorer performance, regardless of the teacher is trained with label smoothing or not.

Table 6: Distillation results using inferior teachers on the long-tailed iNaturalist 2019 (Van Horn et al., 2018) .

| iNaturalist 2019 (Long-tailed) | | | | |
|---|---|---|---|---|
| Teacher | w/ LS | Acc. (Top1/Top3) | Student | Acc. (Top1/Top3) |
| ResNet-50 | ✘ | **73.729/88.845** | ResNet-18 | **67.756/84.521** |
| | | | ResNet-50 | **74.125/88.944** |
| | ✔ | 72.904/87.954 | ResNet-18 | 67.228/84.587 |
| | | | ResNet-50 | 72.838/88.152 |
| ResNet-101 | ✘ | **74.389/88.416** | ResNet-50 | **73.894/89.142** |
| | | | ResNet-101 | **74.488/89.109** |
| | ✔ | 73.597/88.779 | ResNet-50 | 72.409/87.657 |
| | | | ResNet-101 | 73.234/88.647 |

## C   A PYTORCH-LIKE CODE FOR CALCULATING STABILITY METRIC

**Algorithm 1** PyTorch-like Code for Calculating Stability Metric.

```
# x, target: input images and the corresponding labels
# single_std: std value for each class
# N: batchsize (we use N=50 on ImageNet for ease of implementation as it has 50 val images
    in each class)
# model_path: model path that you want to calculate stability

# load model
checkpoint = torch.load(model_path)
model.load_state_dict(checkpoint['state_dict']) # initialize
for (x, target) in loader: # load a minibatch x with N samples (we choose N=50 for ImageNet)
    # make sure shuffle = False in dataloader

    # compute output
    output = model(x)

    # compute softmax probabilities
    softmax_p = nn.functional.softmax(output, dim=1)
    predict = softmax_p[:, target[0]] # obtain target probability

    # compute stability
    single_std = predict.std(dim=0)
    all_std += single_std # aggregate all std values from each class

return 1 - all_std/1000.0 # return computed stability
```

# D  TWO DETAILED TABLES OF RESULTS

Table 7: Image classification results on ImageNet-1K, CUB200-2011 and iMaterialist product recognition. The teacher networks with label smoothing are denoted by "✔". We report average over 3 runs for ResNet-50 as the teacher network on both teacher training and knowledge distillation.

| *ImageNet-1K (Standard):* | | | | |
|---|---|---|---|---|
| Teacher | w/ LS | Acc. (Top1/Top5) | Student | Acc. (Top1/Top5) |
| ResNet-50 | ✘ | $76.056 \pm 0.119/92.791 \pm 0.106$ | ResNet-18 | $71.425 \pm 0.038/90.185 \pm 0.075$ |
| | | | ResNet-50 | $76.325 \pm 0.068/92.984 \pm 0.043$ |
| | ✔ | $\mathbf{76.128 \pm 0.069/92.977 \pm 0.030}$ | ResNet-18 | $\mathbf{71.816 \pm 0.017/90.466 \pm 0.074}$ |
| | | | ResNet-50 | $\mathbf{77.052 \pm 0.030/93.376 \pm 0.015}$ |
| ResNet-101 | ✘ | 77.374/93.546 | ResNet-50 | 77.428/93.712 |
| | | | ResNet-101 | 78.270/94.152 |
| | ✔ | **77.836/93.662** | ResNet-50 | **77.624/93.862** |
| | | | ResNet-101 | **78.476/94.008** |

| *CUB200-2011 (Fine-grained):* | | | | |
|---|---|---|---|---|
| Teacher | w/ LS | Acc. (Top1/Top5) | Student | Acc. (Top1/Top5) |
| ResNet-50 | ✘ | $79.931 \pm 0.037/94.370 \pm 0.064$ | ResNet-18 | $77.116 \pm 0.086/93.241 \pm 0.108$ |
| | | | ResNet-50 | $80.910 \pm 0.033/94.738 \pm 0.114$ |
| | ✔ | $\mathbf{81.497 \pm 0.035/95.043 \pm 0.112}$ | ResNet-18 | $\mathbf{78.382 \pm 0.099/93.621 \pm 0.120}$ |
| | | | ResNet-50 | $\mathbf{82.355 \pm 0.050/95.440 \pm 0.075}$ |
| ResNet-101 | ✘ | 80.380/94.491 | ResNet-50 | 81.261/94.905 |
| | | | ResNet-101 | 81.572/**95.371** |
| | ✔ | **82.332/94.970** | ResNet-50 | **82.263/95.320** |
| | | | ResNet-101 | **82.522**/95.199 |

| *iMaterialist-2019_P (Noisy):* | | | | |
|---|---|---|---|---|
| Teacher | w/ LS | Acc. (Top1/Top3) | Student | Acc. (Top1/Top3) |
| ResNet-50 | ✘ | 66.241/91.015 | ResNet-18 | 65.250/90.243 |
| | | | ResNet-50 | 67.420/92.155 |
| | ✔ | **66.825/91.669** | ResNet-18 | **65.359/90.530** |
| | | | ResNet-50 | **67.528/92.551** |
| ResNet-101 | ✘ | 66.726/91.263 | ResNet-50 | 67.905/92.481 |
| | | | ResNet-101 | 68.281/92.580 |
| | ✔ | **67.370/91.877** | ResNet-50 | **67.925/92.789** |
| | | | ResNet-101 | **68.618/92.907** |

# E  RESULTS OF KNOWLEDGE DISTILLATION WITH HARD AND SOFT LABELS

Here we verify whether label smoothing is still effective when adopting hard labels in knowledge distillation. As we mentioned above, the traditional distillation loss can be formulated as

Table 8: ImageNet results on the same student structure with different teachers. "Long" indicates we train with more budget (160 epochs), the default is 90. "R50 and R18" are ResNet-50/18, respectively.

| Teacher (same arch) | Acc. (Top-1) | Student | Acc. (Top-1) | Teacher (different archs) | Acc. (Top-1) | Student | Acc. (Top-1) |
|---|---|---|---|---|---|---|---|
| R50 | 76.056 | R18 | 71.478 | MobileNet V2 | 71.878 | R18 | 70.054 |
| R50+LS | 76.212 | R18 | 71.816 | DenseNet-121 | 74.894 | R18 | 70.922 |
| R50+LS+Long | 77.106 | R18 | 72.024 | Wide ResNet-50-2 | 77.808 | R18 | 72.232 |
| R50+LS+Long+CutMix | 77.274 | R18 | 72.172 | ResNeXt-101-32x8d | 79.698 | R18 | 72.412 |

Table 9: Distillation results with different ratios of the combination with hard labels and soft labels. The teacher network is ResNet-50 and the student is ResNet-18.

| | | ImageNet |
|---|---|---|
| Ratio (hard label – soft label) | w/ LS | Acc. (Top1/Top5) |
| 0.3 – 0.5 | ✗ | 71.592/90.386 |
| | ✔ | **71.752/90.412** |
| 0.5 – 0.5 | ✗ | 71.484/90.218 |
| | ✔ | **71.748/90.454** |
| 0.7 – 0.3 | ✗ | 71.164/90.196 |
| | ✔ | **71.314/90.200** |

$\lambda \mathcal{H}(\boldsymbol{p}^{\mathcal{S}_{\mathrm{w}}}, \boldsymbol{y}) + (1 - \lambda)\mathcal{H}(\boldsymbol{p}^{\mathcal{S}_{\mathrm{w}}}/\mathcal{T}, \boldsymbol{p}^{\mathcal{T}_{\mathrm{w}}}/\mathcal{T})$. We use ResNet-50 as the teacher and ResNet-18 as the student and conduct experiments with three different ratios of $\lambda$: 0.3, 0.5, 0.7. Our Top-1/5 results are given in Table 9. We can see the teacher networks with label smoothing still distills better students than the teacher without label smoothing. Also, with a higher probability of hard labels, the performance declines.

## F  MINOR PROBABILITIES IN TEACHER NETWORK

The visualization of minor probabilities is shown in Fig. 4 (2), we have two interesting observations in it: 1) In the enlarged area of Fig. 4 (2), we can observe that several classes have values are close to zero, this phenomenon means in ImageNet dataset, there are several categories that are very unique to other classes and the model will barely predict other classes to them; 2) We can also observe more "red" areas gather at the bottom of the bars. It reflects that the probabilities from the model trained with label smoothing will be assigned to more classes with smaller values, this is also consistent with what the label smoothing operation actually does.

## G  SUPPLEMENTARY METRICS OF INTER-CLASS/ENTIRE VARIATIONS

To better understand the behavior of erased information by label smoothing across different classes, we also introduce the inter-class stability metric as follows:

$$\mathcal{S}_{\text{Stability}}^{\text{inter}} = 1 - \frac{1}{K}\sum_{c=1}^{K}(||\overline{\boldsymbol{p}}_{\{c\}}^{\mathcal{T}_{\mathrm{w}}} - \overline{\boldsymbol{p}}_{\{all\}}^{\mathcal{T}_{\mathrm{w}}}||^2) \tag{5}$$

where $K$ is the number of classes. $\overline{\boldsymbol{p}}_{\{all\}}^{\mathcal{T}_{\mathrm{w}}}$ is average of probability across the entire dataset. $\overline{\boldsymbol{p}}_{\{c\}}^{\mathcal{T}_{\mathrm{w}}}$ is the mean of $\boldsymbol{p}^{\mathcal{T}_{\mathrm{w}}}$ in class $c$. $\mathcal{S}_{\text{Stability}}^{\text{inter}}$ can be regarded as a supplement for our intra-class stability metric to measure the inter-class stability. The average probability within each class $\overline{\boldsymbol{p}}_{\{c\}}^{\mathcal{T}_{\mathrm{w}}}$ is defined as:

$$\overline{\boldsymbol{p}}_{\{c\}}^{\mathcal{T}_{\mathrm{w}}} = \frac{1}{n_c}\sum_{i=1}^{n_c}\boldsymbol{p}_{\{i,c\}}^{\mathcal{T}_{\mathrm{w}}} \tag{6}$$

where $i$ is the index of images and $n_c$ is the #image in class $c$. The average probability across entire dataset $\overline{\boldsymbol{p}}_{\{all\}}^{\mathcal{T}_{\mathrm{w}}}$ is defined as:

$$\overline{\boldsymbol{p}}_{\{all\}}^{\mathcal{T}_{\mathrm{w}}} = \frac{1}{N}\sum_{i=1}^{N}\boldsymbol{p}_{\{i\}}^{\mathcal{T}_{\mathrm{w}}} \tag{7}$$

Table 10: Accuracy and inter-class stability results with and without label smoothing on ImageNet-1K. Here we show (1-$\mathcal{S}^{\text{inter}}_{\text{Stability}}$), which denotes the aggregated inter-class variance (the lower the better). Red numbers are the quantitative values of the erased information by label smoothing.

| Netowrks | Acc. (%) w/o LS | (1-$\mathcal{S}^{\text{inter}}_{\text{Stability}}$) w/o LS | Acc. (%) w/ LS | (1-$\mathcal{S}^{\text{inter}}_{\text{Stability}}$) w/ LS |
|---|---|---|---|---|
| ResNet-18 (He et al., 2016) | 69.758/89.078 | 0.1858 | **69.774/89.122** | **0.1724 (-0.0134)** |
| ResNet-50 (He et al., 2016) | 75.888/92.642 | 0.1733 | **76.130/92.936** | **0.1610 (-0.0123)** |
| ResNet-101 (He et al., 2016) | 77.374/93.546 | 0.1671 | **77.726/93.830** | **0.1646 (-0.0025)** |
| MobileNet v2 (Sandler et al., 2018) | 71.878/90.286 | 0.1797 | – | **0.1726 (-0.0071)** |
| DenseNet121 (Huang et al., 2017) | 74.434/91.972 | 0.1763 | – | **0.1666 (-0.0097)** |
| ResNeXt50 32×4d (Xie et al., 2017) | 77.618/93.698 | **0.1658** | **77.774/93.642** | 0.1729 (+0.0071) |
| Wide ResNet50 (Zagoruyko & Komodakis, 2016) | **78.468/94.086** | **0.1602** | 77.808/93.682 | 0.1688 (+0.0086) |
| ResNeXt101 32×8d (Xie et al., 2017) | 79.312/94.526 | **0.1596** | **79.698/94.768** | 0.1677 (+0.0081) |

where $N$ is the number of samples in the entire dataset. This metric utilizes the probabilities of inter-class variance to measure the stability of a teacher's prediction. The results on various network architectures are shown in Table 10.

