# OpenReview forum: "Is Label Smoothing Truly Incompatible with Knowledge Distillation: An Empirical Study"
_ICLR.cc/2021/Conference — ICLR 2021 Poster_

### Official Review · AnonReviewer2 · 2020-10-26
**Official Blind Review #2**

**Rating:** 8
**Confidence:** 4

**Review:**

Recent literature proposed that even label smoothing improves the teacher model, it will hurt the distillation training of student models due to the information erasing. Although this idea dominated more and more literature, this paper argued that this observation is not entirely correct. In order to clarify this idea, the paper systematically discussed the correlation between knowledge distillation and label smoothing. Comprehensive experiments well support the claims in this paper, i.e. label smoothing is compatible with knowledge distillation. The correlation between label smoothing and knowledge distillation remains an open question to date, and this paper made a breakthrough regarding this question. Besides the main purpose (clarify previous ideas), this paper also provided multiple interesting empirical conclusions, e.g. a better teacher always leads to a better student by producing more informative distillation labels, the distillation itself can provide enough regularization for training and the hard-label classification loss is no more needed.

To conclude, the paper overturns the previous perspective with convincing explanations, discussions, and experimental results. Several empirical discoveries are introduced, which are expected to have high impacts on the tasks of knowledge distillation. The major contributions of this paper can be concluded as:
1) The paper empirically confirmed that label smoothing is well compatible with knowledge distillation, overturning previous dominant ideas. This is an important finding because it can prevent subsequent research from being misled.
2) It further explained the phenomenon of relative informative erasing, which only happens on the semantically different classes. Thus previous lopsided ideas (label smoothing hurts knowledge distillation) can be well explained.
3) The paper claimed that the dominating factor in knowledge distillation is the performance of the teacher and further proposed a stability metric to measure the quality of supervision. This metric is crucial in the tasks of knowledge distillation since it provides a simpler and faster way to measure distillation quality. The paper also claimed that the distillation loss itself can provide enough regularization, which also inspires me a lot.

It's quite a good empirical paper and I really enjoy reading it. I think there's no significant weakness on it, so I recommend a clear acceptance for this paper.

---

> ### Author Response · Authors · 2020-11-21
> **Response to Reviewer 2**
>
> We thank the reviewer for the positive feedback, and recognizing our conceptual novelty and contributions. If you have further concerned question, it is our pleasure to give you an answer.

---

### Official Review · AnonReviewer4 · 2020-10-27
**Review of AnonReviewer4**

**Rating:** 6
**Confidence:** 3

**Review:**

The paper empirically discusses the relationship between Label Smoothing (LS) and Knowledge Distillation (KD). It designs a  stability metric to measure the degree of erasing information and finds that LS can be compatible with knowledge distillation except in long-tailed distribution and increased number of classes.

Strengths:

1. Adopting stability metric to measure the degree of erasing information quantitatively is straightforward and effective.
2. Extensive experiments from image classification to NMT are conducted to reveal the relationship between LS and KD and validate the idea of the work.

Weaknesses:
1. Even this work discusses the detailed relationship between LS and KD, but for me, the finding of this work is not enough to reach the bar of the top-tier conferences.
2. The paper has many claims (Bold or Italic), some of that are well-known by the KD community, but too many claims make readers lose what is the paper's key insight.
3. I am still confused that how do the finds in the paper can guide the KD community?  I think "better supervision is crucial for distillation" or "better teachers usually distill better students" should be a well-known and practical skill for KD, and I think many researchers in the KD community don't agree with the conclusion that "a teacher with better accuracy is not necessary to distill a better student" in [1].

[1]. When does label smoothing help? In Advances in Neural Information Processing Systems, Muller, et al.

After reading the response from the authors, I would like to increase my rating to 6: Marginally above acceptance threshold.

---

> ### Author Response · Authors · 2020-11-21
> **Response to Reviewer 4**
>
> We sincerely appreciate the reviewer for correctly recognizing some issues of the many claims (Bold or Italic). We would like to, however, rebut some points that are critical to understand our contributions:
>
> &nbsp;
> >1. Even this work discusses the detailed relationship between LS and KD, but for me, the finding of this work is not enough to reach the bar of the top-tier conferences.
>
> We respect the reviewer’s opinion on the bar of the top-tier conferences, but we would like to rebut that understanding the mechanisms behind these techniques is as valuable as proposing new methods to improve performance on benchmarks. In this study, we clarified, rectified and supplemented several incorrect statements from previous studies, and further provided practical guidelines for further employing our discoveries. We believe such contributions are significant and valuable to the community.
>
> &nbsp;
> >2. The paper has many claims (Bold or Italic), some of that are well-known by the KD community, but too many claims make readers lose what is the paper's key insight.
>
> Thanks for the suggestion. We have removed some bold or italic statements, which are not core claims of the paper, to minimize unnecessary distraction. Please refer to our revised manuscript.
>
> &nbsp;
> >3. I am still confused that how do the finds in the paper can guide the KD community? I think "better supervision is crucial for distillation" or "better teachers usually distill better students" should be a well-known and practical skill for KD, and I think many researchers in the KD community don't agree with the conclusion that "a teacher with better accuracy is not necessary to distill a better student" in [1].
> [1]. When does label smoothing help? In Advances in Neural Information Processing Systems, 2019. Muller, et al.
>
> First, we would like to emphasize that these statements are not well-known in KD community since there are many different settings and arguments in this task. Moreover, our work is not only for readers to better understand knowledge distillation, but also the label smoothing. We reveal the facts that label smoothing is compatible with knowledge distillation and show the circumstances in which label smoothing could lose its effectiveness. Also, we provided novel analysis on how label smoothing helps knowledge distillation. Furthermore, it can achieve greater gains if our proposed guidelines are followed. For example, if the distribution of the dataset is balanced with a small number of classes, label smoothing can achieve better performance and thus should be adopted. On the other hand, if the distribution is long-tailed, label smoothing may be harmful. On CUB200-2011, which contains 200 classes, we observed that adopting label smoothing can generally gain 1.5~2% improvement, while there is no improvement when the class distribution is long-tailed. We believe these practical principles are valuable to the community to evaluate if and when to utilize label smoothing and knowledge distillation under different circumstances, which have not been fully explored before.
>
> Further, we understand that the reviewer may not be in full agreement with some statements in [1]. However, we think [1] is still a valuable study for understanding the mechanisms of label smoothing and knowledge distillation. The proposed visualization scheme and analysis procedure in [1] are insightful, even though some statements may indeed be incorrect. Therefore, our work becomes especially crucial to help correct these misleading claims and draw accurate conclusions, preventing the subsequent research from being misled. We are confident that this study offers critical clarification of [1] to the community for better understanding of the underlying characteristics of label smoothing and knowledge distillation.

---

### Official Review · AnonReviewer3 · 2020-10-29
**Interesting paper, but need some clarification**

**Rating:** 6
**Confidence:** 5

**Review:**

**Paper Summary**
The authors re-analyze and re-confirm the relationship between label-smoothing and knowledge distillation, which is firstly argued by Muller et al. (“When Does Label Smoothing Help?“, NerurIPS 2019.). This paper shows that the previous argument, "label smoothing is not helpful for knowledge distillation", does not always hold, and carefully re-visits the missing points of the previous analysis by Muller et al. Based on this analysis, label smoothing can be helpful for knowledge distillation and can be explained using the intra-class variation and between-class distance within similar classes. The authors have empirically verified the arguments of the paper with various experiments.

**Pros**
1. Introduced an interesting analysis for improved knowledge distillation via label smoothing.
2. The paper is well written and the contributions are clearly explained by comparing against the previous work (Muller et al.)

**Cons**
1. The main analysis is based on the original knowledge distillation paper (Hinton et al., 2015), therefore, it seems to be difficult to apply to the recent knowledge distillation. For example, the proposed analysis cannot explain the recent works (Relational Knowledge Distillation (CVPR 2019), or Contrastive Representation Distillation (ICLR 2020) which are based on the "relation".
2. The stability metric $S_{stability}$ is based on the intra-class variation, but the sample variation of the entire dataset is not considered. Since the intra-class variation is proportional to the variation of the entire samples, the increase of the stability metric might come from the decrease of the entire samples' variation. For example, LDA (linear discriminant analysis) utilizes both intra- and between-class variation. Thus the stability metric needs to be improved and the experiment's scores need to be measured again.
3. Indeed, the knowledge distillation was originally proposed using cross-entropy [1][2], so section 4 (explaining that distillation loss is the same as cross-entropy loss) is not a new observation and it is not necessary for the paper's flow.
[1] Distilling the Knowledge in a Neural Network (NeurIPS workshop 2015)
[2] Fitnets : FITNETS: HINTS FOR THIN DEEP NETS (ICLR 2015)
4. Figure 4 needs to be improved. It is hard to see and capture the meaning of "other soft predictions" in the figure. It would be nice to improve the figure 4 so that it clearly expresses what it is intended.
5. In appendix D, does the "loader" utilize standard ImageNet data augmentation (e.g., random-resize-cropping)?

**Comments after the author response**
- The authors have answered all my questions and they revised the manuscript as well, so I will keep my initial rating (weak accept).

---

> ### Author Response · Authors · 2020-11-21
> **Response to Reviewer 3 (3/3)**
>
> &nbsp;
> >3. Indeed, the knowledge distillation was originally proposed using cross-entropy [1][2], so section 4 (explaining that distillation loss is the same as cross-entropy loss) is not a new observation and it is not necessary for the paper's flow. [1] Distilling the Knowledge in a Neural Network (NeurIPS workshop 2015) [2] Fitnets : FITNETS: HINTS FOR THIN DEEP NETS (ICLR 2015)
>
> Sec. 4 aims to demonstrate that solely based on soft labels, we can already achieve competitive results, implying that the distillation loss is mathematically equivalent to the cross-entropy loss on soft labels only. Whereas, in the two papers mentioned by the reviewer, the distillation loss includes both hard labels and soft labels of the cross-entropy loss, which is slightly different from our observation. We have clarified this point in the revised manuscript.
>
> &nbsp;
> >4. Figure 4 needs to be improved. It is hard to see and capture the meaning of "other soft predictions" in the figure. It would be nice to improve the figure 4 so that it clearly expresses what it is intended.
>
> Thanks for this suggestion. We have separated the major and minor/small probabilities in Fig. 4 with two subfigures. Some discussions are also given in the Appendix F of our revision.
>
> &nbsp;
> >5. In appendix D, does the "loader" utilize standard ImageNet data augmentation (e.g., random-resize-cropping)?
>
> Random-resize-cropping is not used as the stability metric is calculated on the validation set (this is mentioned in line 3 of Algorithm 1), so the only strategy we used for calculating this metric is to resize the image to 256x256 then center-crop it to 224x224, following the ImageNet validation protocol.

---

> ### Author Response · Authors · 2020-11-21
> **Response to Reviewer 3 (2/3)**
>
> &nbsp;
> >2. The stability metric $S_{stability}$ is based on the intra-class variation, but the sample variation of the entire dataset is not considered. Since the intra-class variation is proportional to the variation of the entire samples, the increase of the stability metric might come from the decrease of the entire samples' variation. For example, LDA (linear discriminant analysis) utilizes both intra- and between-class variation. Thus the stability metric needs to be improved and the experiment's scores need to be measured again.
>
> As the "erasing information" effect by label smoothing mainly occurs within each class, so the intra-class variation metric is basically qualified to measure the erased degree. We certainly agree that monitoring the inter-class variation is also crucial to better understand the effects of erased information by label smoothing across different classes. We provide such results of inter-class stability below following the formulation:
> $$\\mathcal{S}^\\textrm{inter}\_{\\textrm{Stability}}=1-\\frac{1}{K}\\sum\_{c=1}^{K}(\|\|  \\overline {p}\_{\\{c\\}}^{\\mathcal T\_\\mathbf{w}}- \\overline {p}\_{\\{{all}\\}}^{\\mathcal T\_\\mathbf{w}}\|\|^{2})$$
> where *K* is the number of classes.
> $\\overline {p}\_{\\{{all}\\}}^{\\mathcal T\_\\mathbf{w}}$ is the average of
> probability across the entire dataset.
> $\\overline {p}\_{\\{c\\}}^{\\mathcal T\_\\mathbf{w}}$ is the mean of
> $\{p}\_{}^{\\mathcal T\_\\mathbf{w}}$ in class *c*.
> $\\mathcal{S}^\\textrm{inter}\_{\\textrm{Stability}}$ utilizes the probabilities of inter-class variance to measure the stability. The average probability within each class
> $\\overline {p}\_{\\{c\\}}^{\\mathcal T\_\\mathbf{w}}$ is defined as:
> $$\\overline {p}\_{\\{c\\}}^{\\mathcal T\_\\mathbf{w}}=\\frac{1}{n\_c}\\sum\_{i=1}^{n\_c}  {p}\_{\\{i,c\\}}^{\\mathcal T\_\\mathbf{w}}$$
> where *i* is the index of images and $n\_c$ is the \#image in
> class *c*. The average probability across entire dataset
> $\\overline {p}\_{\\{{all}\\}}^{\\mathcal T\_\\mathbf{w}}$ is defined
> as:
> $$\\overline {p}\_{\\{{all}\\}}^{\\mathcal T\_\\mathbf{w}}=\\frac{1}{N}\\sum\_{i=1}^{N}  {p}\_{\\{i\\}}^{\\mathcal T\_\\mathbf{w}}$$
> where *N* is the number of samples in the entire dataset.
>
> Generally, the trend of inter-class variation is consistent with the intra-class metric but for some architectures the variation has increased. For example, the intra-class stability of ResNeXt50 32x4d increases while the inter-class variation also increases. This phenomenon supports our argument that the semantically similar classes will have larger distances with label smoothing. Hence, for the inter-class variation, we believe studying the semantically similar and dissimilar classes as we did in this paper (since particular classes’ variations will be averaged) may be more insightful for understanding the “erasing information” effect. We have included these new results and some discussions in Appendix G.
>
> | |  Acc. (%) w/o LS| (1-$\\mathcal{S}^\\textrm{intra}\_{\\textrm{Stability}}$) w/o LS| (1-$\\mathcal{S}^\\textrm{inter}\_{\\textrm{Stability}}$) w/o LS | Acc. w/ LS (%) |  (1-$\\mathcal{S}^\\textrm{intra}\_{\\textrm{Stability}}$) w/ LS   |     (1-$\\mathcal{S}^\\textrm{inter}\_{\\textrm{Stability}}$) w/ LS  |
> | :---: | :--: | :-------: | :------:| :------: | :------: | :------: |
> |ResNet-18     |    69.758/89.078   | 0.3359  |  0.1858  |  69.774/89.122  |  0.3358   |  0.1724  |
> |ResNet-50  |         75.888/92.642  |    0.3217 |   0.1733  |  76.130/92.936  |  0.3106   |   0.1610 |
> |ResNet-101   | 77.374/93.546   |  0.3185 | 0.1671   | 77.726/93.830 |  0.3070    |     0.1646   |
> |MobileNet v2    |    71.878/90.286   |    0.3341 |    0.1797  |      –    |                  –     |            0.1726  |
> |DenseNet121    |     74.434/91.972 |      0.3243 |   0.1763   |    –     |                 –       |             0.1666   |
> |ResNeXt50 32×4d     |     77.618/93.698  |     0.3229  |   0.1658   |  77.774/93.642  |    0.3182    |           0.1729   |
> |Wide ResNet50  |       78.468/94.086   |      0.3201 |  0.1602  |  77.808/93.682 |    0.3155   |          0.1688   |
> |ResNeXt101 32×8d     |     79.312/94.526  |     0.3177  | 0.1596  | 79.698/94.768 |  0.3116     |         0.1677   |

---

> ### Author Response · Authors · 2020-11-21
> **Response to Reviewer 3 (1/3)**
>
> We thank you for taking the time to review our paper and we appreciate the valuable feedback. Please see our responses and clarifications for your questions below. We have posted a revision of the paper and will continue updating it during the discussion period.
>
> &nbsp;
> >1. The main analysis is based on the original knowledge distillation paper (Hinton et al., 2015), therefore, it seems to be difficult to apply to the recent knowledge distillation. For example, the proposed analysis cannot explain the recent works (Relational Knowledge Distillation (CVPR 2019), or Contrastive Representation Distillation (ICLR 2020) which are based on the "relation".
>
> Essentially, the relational knowledge distillation explores the angles and distances between classes. The teacher with label smoothing can enlarge the distance between the semantically similar classes, which may also help the relational knowledge distillation.
>
> Contrastive representation distillation explores the structural knowledge of the teacher network based on the contrastive learning. Basically, contrastive learning utilizes a similarity or matching function to measure the similarity of two representations, thus our analysis on semantically similar or dissimilar classes is still suitable for such an objective. We can explore strategies of identifying optimal positive and negative pairs for contrastive distillation. We believe our analysis on the representation of distances between semantically similar or dissimilar classes could potentially help better understand these relation-based distillation methods.
>
> We haven conducted an exemplar experiment to evaluate the effect of label smoothing on Relational Knowledge Distillation. In the experiment, we simply replaced the teachers from metric learning (triplet loss) to our cross-entropy with or without label smoothing and adopted the RKD’s distillation protocol. The following summarizes the results on CUB200-2011 with 200-dimensional embedding:
>
> Teacher (ResNet-50)    Student (ResNet-18):
>
>         w/ label smoothing    Train Recall: 0.9023    Eval Recall: 0.5939
>
>         w/o label smoothing   Train Recall: 0.8712    Eval Recall: 0.5133
>
> These results demonstrate that label smoothing is also helpful in such a relation-based framework.

---

### Official Review · AnonReviewer1 · 2020-11-06

**Rating:** 6
**Confidence:** 3

**Review:**

This paper is mainly based on the prior work by Muller et al., which suggests that label smoothing is incompatible with knowledge distillation. Firstly, this paper provides an explanation of this incompatibility---label smooth tends to erase relative information among different classes, and provide a way to qualitatively measure the degree of erased information. Then, this paper argues that label smoothing actually is compatible with knowledge distillation, and show several empirical results as evidence. Lastly, this paper suggests that the performance of the teacher model is a more directly related factor for determining the performance of the student model.


Pros:

(1) This paper performs a set of careful diagnoses on showing the effects of information erasing (caused by using label smoothing) at the category-level, and observes an interesting phenomenon: erasing relative information only cause negative effects to semantically different classes, but will help the classification for semantically similar classes. Both qualitative and quantitative evidence is provided to confirm this observation.

(2) A simple and novel metric is proposed to facilitate the measurement of the degree of erased information.

(3) Extensive experiments on the image classification task and the neural machine translation task are provided to confirm that label smoothing is indeed compatible with the knowledge distillation framework.



Cons (please address them during the rebuttal):

(1) The presentation of this paper needs to be improved. In all abstract, introduction and conclusion sections, this paper highlights that "we broadly discuss several circumstances wherein label smoothing will indeed lose its effectiveness". Nonetheless, the reviewer CANNOT find any related discussions in the main paper. The only related discussion is provided in the appendix. The reviewer does not think it is a good way for presenting the paper, as the appendix is mainly used for explaining some not very important details. Putting the entire discussion of an important contribution of this paper in the appendix is inappropriate.

(2) Though the reviewer agrees that removing the hard label part in knowledge distillation can facilitate the analysis of this paper, the authors should also provide a brief discussion on whether adding this hard label part back will still lead to the same conclusions. For example, in table 2, with the hard label part, if the teacher model with label smoothing can still help the student model.

(3) One minor question is that, as shown in Table 1, ResNet-50+long substantially outperforms ResNet-50 in terms of accuracy, but their stability measurements are nearly the same. Can the authors provide any explanation of this "counter-intuitive" phenomenon?

---

> ### Author Response · Authors · 2020-11-21
> **Response to Reviewer 1**
>
> We thank you for taking the time to review our paper and we appreciate the valuable comments. Please see our responses and clarifications for your questions below. We have posted a revision of the paper and will continue updating it during the discussion period.
>
> &nbsp;
> >(1) The presentation of this paper needs to be improved. In all abstract, introduction and conclusion sections, this paper highlights that "we broadly discuss several circumstances wherein label smoothing will indeed lose its effectiveness". Nonetheless, the reviewer CANNOT find any related discussions in the main paper. The only related discussion is provided in the appendix. The reviewer does not think it is a good way for presenting the paper, as the appendix is mainly used for explaining some not very important details. Putting the entire discussion of an important contribution of this paper in the appendix is inappropriate.
>
> Thanks for pointing out this. We have moved Section “What Circumstances Indeed Will Make LS Less Effective” from the appendix to Sec. 7 of the main paper, to make the main paper more self-contained.
>
> &nbsp;
> >(2) Though the reviewer agrees that removing the hard label part in knowledge distillation can facilitate the analysis of this paper, the authors should also provide a brief discussion on whether adding this hard label part back will still lead to the same conclusions. For example, in table 2, with the hard label part, if the teacher model with label smoothing can still help the student model.
>
> We agree that further analysis of combining the soft and hard labels would help gain further insights and offer additional evidences supporting the conclusion. We thus conducted experiments, using ResNet-50 as the teacher and ResNet-18 as the student, with three different ratios (0.3, 0.5, 0.7). The following are our Top-1/5 results:
>
> Hard label (0.3) + Soft label (0.7) :  w/o label smoothing: 71.592/90.386 &nbsp;   w/ label smoothing: 71.752/90.412
>
> Hard label (0.5) + Soft label (0.5) :  w/o label smoothing: 71.484/90.218  &nbsp;  w/ label smoothing: 71.748/90.454
>
> Hard label (0.7) + Soft label (0.3) :  w/o label smoothing: 71.164/90.196  &nbsp;   w/ label smoothing: 71.314/90.200
>
> The results indicate that the teacher networks with label smoothing still distill better students than the teacher without label smoothing. Also, with a higher ratio of hard labels, the performance declines. We have added some discussions about this analysis in Sec. 5 of the main paper, and also included these results in Appendix E.
>
> &nbsp;
> >(3) One minor question is that, as shown in Table 1, ResNet-50+long substantially outperforms ResNet-50 in terms of accuracy, but their stability measurements are nearly the same. Can the authors provide any explanation of this "counter-intuitive" phenomenon?
>
> We conjecture this phenomenon is related to the model confidence degree of predictions, also called confidence calibration, i.e., predicting the probability estimation of the true correctness. By definition, accuracy only measures the correctness of the highest prediction, without taking into account the confidence degree of predictions.  One observation from Table 1 is that some regularization methods, such as label smoothing and CutMix, can dramatically increase the stability. We believe these techniques can make the probability better calibrated, in turn improving the stability, while long training cannot achieve this purpose. We further validated this conjecture by testing the distilled ResNet-50 (trained with dynamic supervision) and observed similar improvement (0.3037).

---

### Comment · ~Xinshao_Wang1 · 2021-07-02
**Not All Knowledge Is Created Equal: Rethinking Knowledge Distillation https://arxiv.org/abs/2106.01489**

Dear all authors:

First, great job, congratulations.

Here, I would like to present some of my personal thoughts to discuss with you and public peers who have similar interests.

1. For the statement that " label smoothing could improve calibration implicitly but will hurt the effectiveness of knowledge distillation (Muller et al., 2019)", my different perspective is that there are **two factors in KD: knowledge source (teacher) and distillation process. These two factors are not independent. That is why a better teacher alone may not improve a student without modifying distillation process, and this is not specific to label smoothing at all.** Please have a look at our recent work for more detailed information: https://arxiv.org/abs/2106.01489

Surprisingly, our very different perspectives lead to the same conclusion: a better teacher improved by label smoothing is not the reason of suppressing knowledge distillation's effectiveness.


2. For the statement that "label smoothing tends to “erase” information contained intra-class across individual examples", is not this the same for different classes as well? Please have a brief look at our recent work for more details about semantic class and class similarity structure defined by label annotations: https://xinshaoamoswang.github.io/blogs/2020-06-07-Progressive-self-label-correction/


3. For the statement "erasing phenomenon can enforce two clusters being away from each other and actually enlarge the central distance of clusters between classes", in other words, "label smoothing forces similar classes to be more far away in the embedding space"?  Therefore, label smoothing helps to distinguish semantically similar classes. I fully agree with this statement and finding.


4. For the increased number of classes, what do you find? Do you mean that label smoothing is not good for large-scale datasets? But label smoothing was initially proposed for ImageNet in: https://arxiv.org/pdf/1512.00567.pdf


5. Following my point #4 above, for More #Class, in Table 4, for different size of training datasets, how is the size of testing datasets?
I am unsure whether the comparison is valid without taking into consideration of the testing data size.
    * 0.88 * 50,000 / 10 (since only 100 classes) = 4,400 examples improved
    * 0.455 * 50000 / 2 (500 classes) = 11,375 examples improved
    * 0.155 * 50000 / 1 (1000 classes) = 7,750 examples improved.
**In other words, the percent measurement is inconsistent with the exact number of examples**


6. Do you have any insights and experimental studies on label noise, or semi-supervised learning in addition to long-tailed distribution?

Thanks very much.
Best wishes.

---

> ### Comment · ~Zhiqiang_Shen1 · 2021-07-02
> **Thanks for sharing the interesting paper.**
>
> Hi Xinshao,
>
> Thanks very much for your shared insights and the interesting work, I will definitely read it. For your mentioned two questions (i) "increased number of classes". The most direct and intuitive discovery we found is that the improvement on CUB200-2011 using label smoothing is significantly better than that on ImageNet-1K (generally, the boost with label smoothing on ImageNet-1K is not stable and heavily associated with the backbone networks you are using), so we conjecture the number of class in the dataset will affect the effectiveness of label smoothing since CUB200-2011 only contains 200 classes but ImageNet-1K has 1000. That's the reason we build curated ImageNets with 100, 500 and 1K classes. Overall, label smoothing is still helpful but becomes less effective from 100 to 1000 classes.
>
> For the size of testing datasets, we use the ImageNe-1K val set for testing while the classes are consistent with the training data, which means, like for ImageNet-100, we select 100 classes and use these classes' training and val images to construct the dataset.
>
>  (ii) "label noise": we found the distillation method together with data augmentations like mixup and cutmix is extremely effective for noisy data since the provided supervision from the teacher model is more accurate than the original ones, especially on such noisy circumstance.
>
> For the "long-tailed distribution", currently, we only observe that the label smoothing on such kind of data is useless or even harmful. Further study will be interesting if could make label smoothing effective again on the long-tailed scenario.
>
> Best,
>
> Zhiqiang

---

> > ### Comment · ~Xinshao_Wang1 · 2021-07-02
> > **increased number of classes versus test data size**
> >
> > Dear Zhiqiang:
> >
> > Great to hear from you. I have updated my questions a bit.
> >
> > Since I do not know what subset of classes are used in your experiments, I assume that the samples per class is balanced in ImageNet. The details are as follows:
> >
> > Following my point #4 above, for More #Class, **in Table 4, for different size of training datasets**, how is the size of testing datasets?
> > I am unsure whether the comparison is valid without taking into consideration of the testing data size.
> >
> > * 0.88 * 50,000 / 10 (since only 100 classes) = 4,400 examples improved
> > * 0.455 * 50000 / 2 (500 classes) = 11,375 examples improved
> > * 0.155 * 50000 / 1 (1000 classes) = 7,750 examples improved.
> >
> > In other words, **the percent measurement is inconsistent with the exact number of examples**
> >
> > This question is just out of my curiosity, and does not affect the good points of this paper.

---

> > > ### Comment · ~Zhiqiang_Shen1 · 2021-07-04
> > > **regarding the test data size**
> > >
> > > Dear Xinshao,
> > >
> > > Thanks very much for the more details of the question. I think this is a very good point. Yes, if considering the number of improved images, the tendency will be different from the absolute accuracy change. Maybe we can normalize the improvement of accuracy based on the scale of different datasets but not sure this is necessary and correct, because the larger-scale dataset also has larger training data, this is the advantage of it over the small datasets.
> > >
> > > Best,
> > >
> > > Zhiqiang

---

> > ### Comment · ~Xinshao_Wang1 · 2021-07-03
> > **Example Weighting could be a better approach for sample imbalance, a.k.a. long-tailed distribution**
> >
> > For the "long-tailed distribution", personally, I think example weighting could be a better and more intuitive strategy for handling sample imbalance. For example, some of our recent work is as follows:
> > * Derivative Manipulation for General Example Weighting: https://arxiv.org/abs/1905.11233
> > * IMAE for Noise-Robust Learning: Mean Absolute Error Does Not Treat Examples Equally and Gradient Magnitude's Variance Matters: https://arxiv.org/abs/1903.12141
> > * Selected work partially impacted by them: https://xinshaoamoswang.github.io/blogs/2020-06-14-Robust-Deep-LearningviaDerivativeManipulationIMAE/#1-selected-work-partially-impacted-by-our-work
> >
> > I believe that we share a lot of research interests, and have worked very hard to amend some prior findings. I really look forward to more discussion with you if possible. If there is a chance to meet in-person in the future, that will be wonderful.

---

> > > ### Comment · ~Zhiqiang_Shen1 · 2021-07-04
> > > **Thanks for sharing**
> > >
> > > I think we will definitely have chances to meet physically in future conferences after the pandemic. I also look forward to meeting you at that time!
> > >
> > > Best,
> > >
> > > Zhiqiang

---

> ### Comment · ~Youcai_Zhang1 · 2021-09-24
> **What knowledge to be selected is important in distillation**
>
> Hi Xinshao,
>
> Your perspective about two factors in KD is quite interesting.
>
> Actually, we proposed Prime-aware Adaptive Distillation~(PAD) in ECCV2020. The main contribution of PAD is the introduction of “adaptive sample weighting” into KD, which is an important problem but does not receive enough attention in KD. PAD explored the problem of "What knowledge to be selected for distillation"  by using the modeling of KD with data uncertainty for the first time.
>
> We hope PAD may be helpful to your work.
>
> Best wishes.
>
> Youcai

---

### Decision · Program_Chairs · 2021-01-07
**Final Decision**

**Decision:**

Accept (Poster)

**Comment:**


This paper studies the effect of label smoothing on knowledge-distillation. A previous work on this topic (Muller et al.) has claimed that label smoothing can hurt the performance of the student model in knowledge-distillation. The rationale behind this argument is that label smoothing erases information encoded in the labels. This work shows that such claimed effect does not necessarily happen. Specifically, by a comprehensive study on image classification, binary neural networks, and neural machine translation, the authors show that label smoothing can be compatible with knowledge distillation. However, they conclude that label smoothing will lose its effectiveness with long-tailed distribution and increased number of classes.

Overall ratings of this paper are all on the positive side, and R2 finding this paper an important step toward understanding the interaction between knowledge-distillation and label smoothing. I concur with the reviewers about the importance of this research direction and I think this submission provides a reasonable empirical evidence to change our earlier perspectives. I recommend accept.

While the paper specifically studies the effect of label smoothing on knowledge-distillation, I think providing a bigger context and reviewing some of the recent demystifying efforts on understanding knowledge-distillation could allow paper to communicate with a broader audience. I hope this can be accommodated in the final version.